# Comprehensive annotations of human herpesvirus 6A and 6B genomes reveal novel and conserved genomic features

Yaara Finkel[1], Dominik Schmiedel[2], Julie Tai-Schmiedel[1], Aharon Nachshon[1], Roni Winkler[1], Martina Dobesova[1], Michal Schwartz[1], Ofer Mandelboim[2], Noam Stern-Ginossar[1]*

[1]Department of Molecular Genetics, Weizmann Institute of Science, Rehovot, Israel; [2]The Lautenberg Center for General and Tumor Immunology, Institute for Medical Research Israel-Canada, The Hebrew University Hadassah Medical School, Jerusalem, Israel

**Abstract** Human herpesvirus-6 (HHV-6) A and B are ubiquitous betaherpesviruses, infecting the majority of the human population. They encompass large genomes and our understanding of their protein coding potential is far from complete. Here, we employ ribosome-profiling and systematic transcript-analysis to experimentally define HHV-6 translation products. We identify hundreds of new open reading frames (ORFs), including upstream ORFs (uORFs) and internal ORFs (iORFs), generating a complete unbiased atlas of HHV-6 proteome. By integrating systematic data from the prototypic betaherpesvirus, human cytomegalovirus, we uncover numerous uORFs and iORFs conserved across betaherpesviruses and we show uORFs are enriched in late viral genes. We identified three highly abundant HHV-6 encoded long non-coding RNAs, one of which generates a non-polyadenylated stable intron appearing to be a conserved feature of betaherpesviruses. Overall, our work reveals the complexity of HHV-6 genomes and highlights novel features conserved between betaherpesviruses, providing a rich resource for future functional studies.

*For correspondence:
noam.stern-ginossar@weizmann.ac.il

**Competing interests:** The authors declare that no competing interests exist.

## Introduction

Human herpesvirus 6 (HHV-6) is a ubiquitous betaherpesvirus. Based on distinct molecular, epidemiological and biological properties, two variants of this virus were declared as two separate, closely related, viral species; HHV-6A and HHV-6B (*Ablashi et al., 2014*; *Forni et al., 2019*; *O'Grady et al., 2016*; *Telford et al., 2018*). While HHV-6A remains poorly epidemiologically characterized, it was suspected to associate with neurodegenerative disease such as Alzheimer's disease (Allnutt et al., under review; *Braun et al., 1997*; *Eimer et al., 2018*; *Leibovitch and Jacobson, 2014*; *Prusty et al., 2018b*; *Readhead et al., 2018*). HHV-6B is known to infect more than 90% of the human population (*Zerr et al., 2005*) and was found to be the causative agent of Roseola Infantum, leading to febrile seizures in more than 10% of acute infections (*De Bolle et al., 2005*; *Hall et al., 1994*; *Yamanishi et al., 1988*). Both HHV-6A and HHV-6B, like all herpesviruses, establish a lifelong latent infection in their hosts (*Kondo et al., 1991*; *Luppi et al., 1999*). HHV-6 latency is established in multiple cell types, where the viral genome is integrated into host chromosomes between telomeres and subtelomeres (*Arbuckle et al., 2013*; *Braun et al., 1997*). Remarkably, in approximately 1% of the population worldwide HHV-6 is integrated in every cell in the body, and inherited, due to integration of the viral genome in germline cells (*Clark, 2016*; *Pellett et al., 2012*). HHV-6 reactivation is a common cause of encephalitis, and has been associated with several diseases including multiple sclerosis, hepatitis, pneumonitis and graft-versus-host disease (*Braun et al., 1997*; *Caselli and Di Luca, 2007*; *De Bolle et al., 2005*).

The genomes of HHV-6A and HHV-6B, similar to those of other herpesviruses, consist of large linear double stranded DNA molecules, 160 kb in length, containing a unique segment flanked by direct repeats (*Lindquester and Pellett, 1991*; *Martin et al., 1991*). The annotation of HHV-6 coding capacity has traditionally relied on open reading frame (ORF)-based analyses using canonical translational start and stop sequences and arbitrary size restriction to demarcate putative protein coding genes, resulting in a list of around 100 ORFs for each virus (*Dominguez et al., 1999*; *Gompels et al., 1995*; *Gravel et al., 2013*). In recent years, genome wide-analysis of herpesviruses using short RNA sequencing (RNA-seq) reads, and recently also direct and long-read RNA-seq revealed very complex transcriptomes (*Balázs et al., 2018*; *Balázs et al., 2017*; *Depledge et al., 2019*; *Gatherer et al., 2011*; *Kara et al., 2019*; *O'Grady et al., 2019*; *O'Grady et al., 2016*; *Tombácz et al., 2017*), and combined with genome-wide mapping of translation, revealed hundreds of new viral ORFs (*Arias et al., 2014*; *Bencun et al., 2018*; *Stern-Ginossar et al., 2012*; *Whisnant et al., 2019*). Specifically for HHV-6, recent work using proteomics, transcriptomics and comparative genomics on HHV-6B enabled re-annotation of several viral gene products (*Greninger et al., 2018*). Taken together, this unforeseen complexity of herpesviruses suggests the current annotations of HHV-6 genomes are likely incomplete.

Here, we apply ribosome profiling (Ribo-seq) and RNA-seq to investigate the genomes of the closely related HHV-6A and HHV-6B. These powerful tools allowed us to accurately determine the translation initiation sites of previously annotated genes, and to identify hundreds of new open reading frames including many upstream ORFs (uORFs) and internal ORFs (iORFs), generating a comprehensive atlas of HHV-6 translation products. Using our RNA-seq data, we were able to map novel splice junctions and to identify novel highly abundant viral long non-coding RNAs. The systematic annotations of two betaherpesviruses together with our previous annotation of the prototypic betaherpesvirus human cytomegalovirus (HCMV) (*Stern-Ginossar et al., 2012*) created for the first time an opportunity to look at functional conservation of some of these features. We found high levels of conservation between HHV-6A and HHV-6B, and in several cases, the newly identified features were also conserved in HCMV. Our results shed light on the complexity of herpesviruses, point to conserved features and can serve as a valuable resource for future studies of these important viruses.

## Results

### Profiling the transcriptome and translatome of HHV-6A and HHV-6B

To capture the full complexity of HHV-6A and HHV-6B genomes, we applied next generation sequencing methods that map genome-wide RNA expression and translation to HSB-2 and Molt-3 cells infected for 72 hr with HHV-6A strain GS and HHV-6B strain Z29, respectively (*Figure 1A*). For each virus we mapped genome-wide translation events by preparing three different ribosome-profiling libraries (Ribo-seq). Two Ribo-seq libraries facilitate mapping of translation initiation sites, by treating cells with lactimidomycin (LTM) or harringtonine (Harr), drugs that inhibit translation initiation in distinct mechanisms and lead to accumulation of ribosomes at translation initiation sites (*Figure 1A* and *Ingolia et al., 2011*; *Lee et al., 2012*). The third Ribo-seq library was prepared from cells treated with the translation elongation inhibitor cycloheximide (CHX), and gives a snap-shot of actively translating ribosomes across the body of the translated ORF (*Figure 1A*). In parallel, we used a tailored RNA-sequencing (RNA-seq) protocol which on top of quantification of RNA levels allows identification of transcription start sites (TSSs) due to a strong overrepresentation of fragments that start at the 5' end of transcripts, as well as detection of polyadenylation sites (*Figure 1A* and *Stern-Ginossar et al., 2012*). The combination of these methods provides accurate mapping of transcription and translation events, as seen in the example of U54 (*Figure 1A*). The different Ribo-seq libraries generate distinct profiles across the coding region, displaying a strong peak at the translation initiation site, which, as expected, is more distinct in the Harr and LTM libraries, while the CHX library provides the distribution of ribosomes across the entire coding region up to the stop codon, and its mapped footprints were enriched in fragments that align to the translated frame (*Figure 1—figure supplement 1–*). These profiles were consistent across coding regions in human genes (*Figure 1B*) and, as expected, the RNA-seq profiles were uniformly distributed across the coding region (*Figure 1C*).

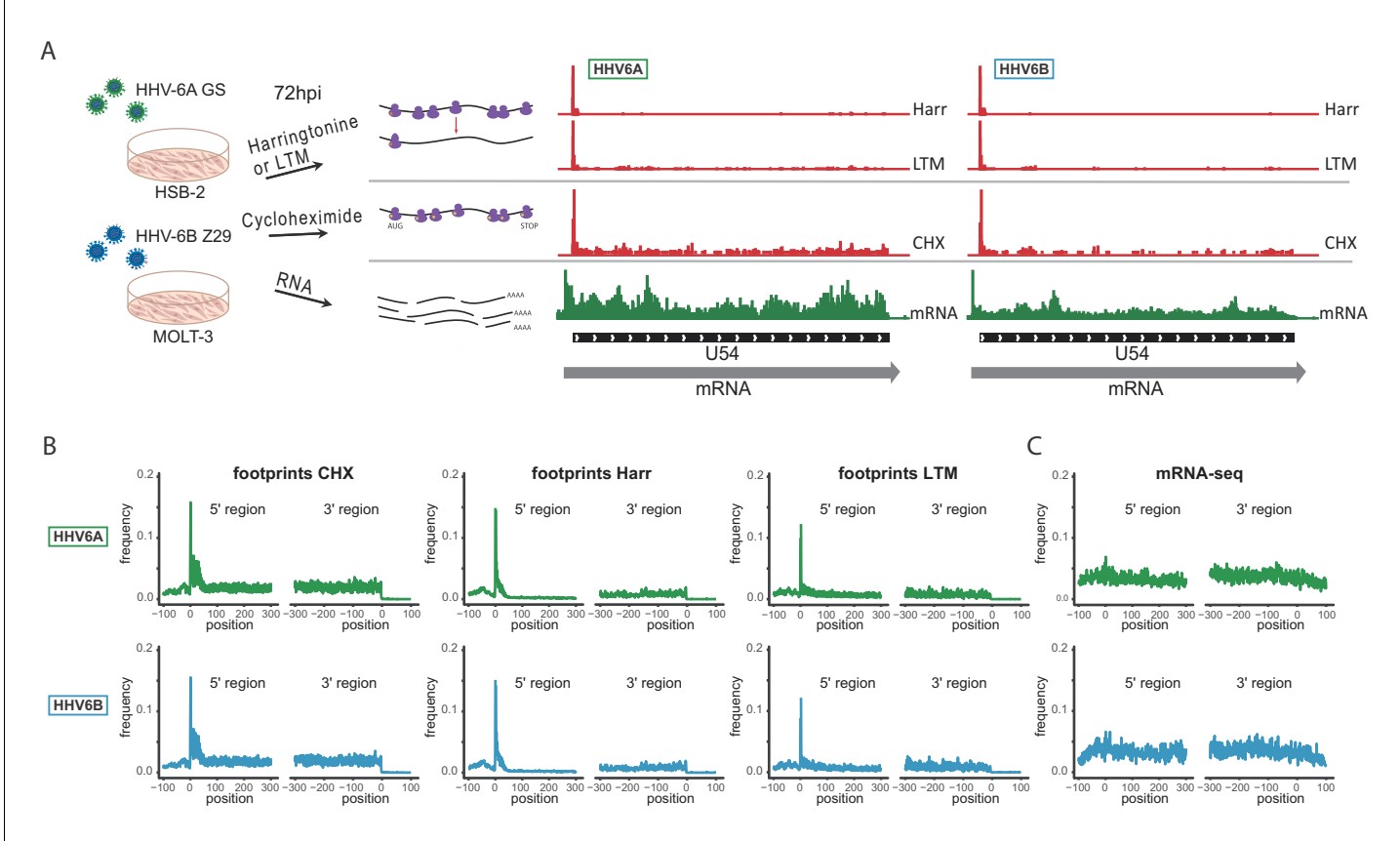

**Figure 1.** Overview of the experimental approach. (**A**) Viral gene expression was analyzed by performing ribosome profiling (red) and initiation enriched RNA-seq (green). HSB-2 cells were infected with HHV-6A strain GS, and MOLT3 cells were infected with HHV-6B strain Z29. Infected cells were harvested at 72 hr post infection (hpi) for RNA-seq, and for ribosome profiling using cycloheximide (CHX) treatment to map overall translation or lactimidomycin (LTM) and Harringtonine (Harr) treatments for mapping translation initiation. (**B-C**) Metagene analysis of the 5' and the 3' regions of human protein coding regions showing the expression profile as measured by the different (**B**) Ribo-seq and (**C**) RNA-seq methods in HHV-6A (green) and HHV-6B (blue) infected cells. The X axis shows the nucleotide position relative to the start or the stop codons.
The online version of this article includes the following figure supplement(s) for figure 1:

**Figure supplement 1.** Reading-frame distribution of Ribo-seq reads.

## Ribo-seq libraries uncover the translation landscape of HHV-6A and HHV-6B

We used the Ribo-seq data to determine translation of viral ORFs. Comparing to previously annotated ORFs, we found many misannotations (10 and 11, in HHV-6A and HHV-6B respectively) and novel un-annotated ORFs (278 and 227, in HHV-6A and HHV-6B respectively). Importantly, many of these new ORFs are conserved between HHV-6A and HHV-6B, validating our approach, and emphasizing the high similarity between these two viruses. One example of misannotation is the U30 gene, an essential viral gene coding for an inner tegument protein (*Nicholas and Martin, 1994*). We found translation of this gene to initiate at an AUG 411 bp downstream of the previously annotated start, in both HHV-6A and HHV-6B, resulting in a 946 amino acid (aa) long protein (*Figure 2A*). Importantly, the new annotations include the C-terminal domain which was shown to interact with the large tegument protein in the HSV-1 homolog (*Richards et al., 2017*).

We identified novel ORFs that are present in both viruses. For example, a short 32 aa ORF was found to initiate upstream of the envelope protein gene U48 (*Figure 2B*). This ORF partially overlaps the U48 gene, making it an upstream overlapping ORF (uoORF). Since uoORFs are known to have repressive regulatory effects conserved across vertebrates (*Johnstone et al., 2016*), this novel ORF likely negatively regulates the translation of U48. We did not observe translation of another downstream ORF that could be positively regulated by this uoORF. The packaging gene U36 is an

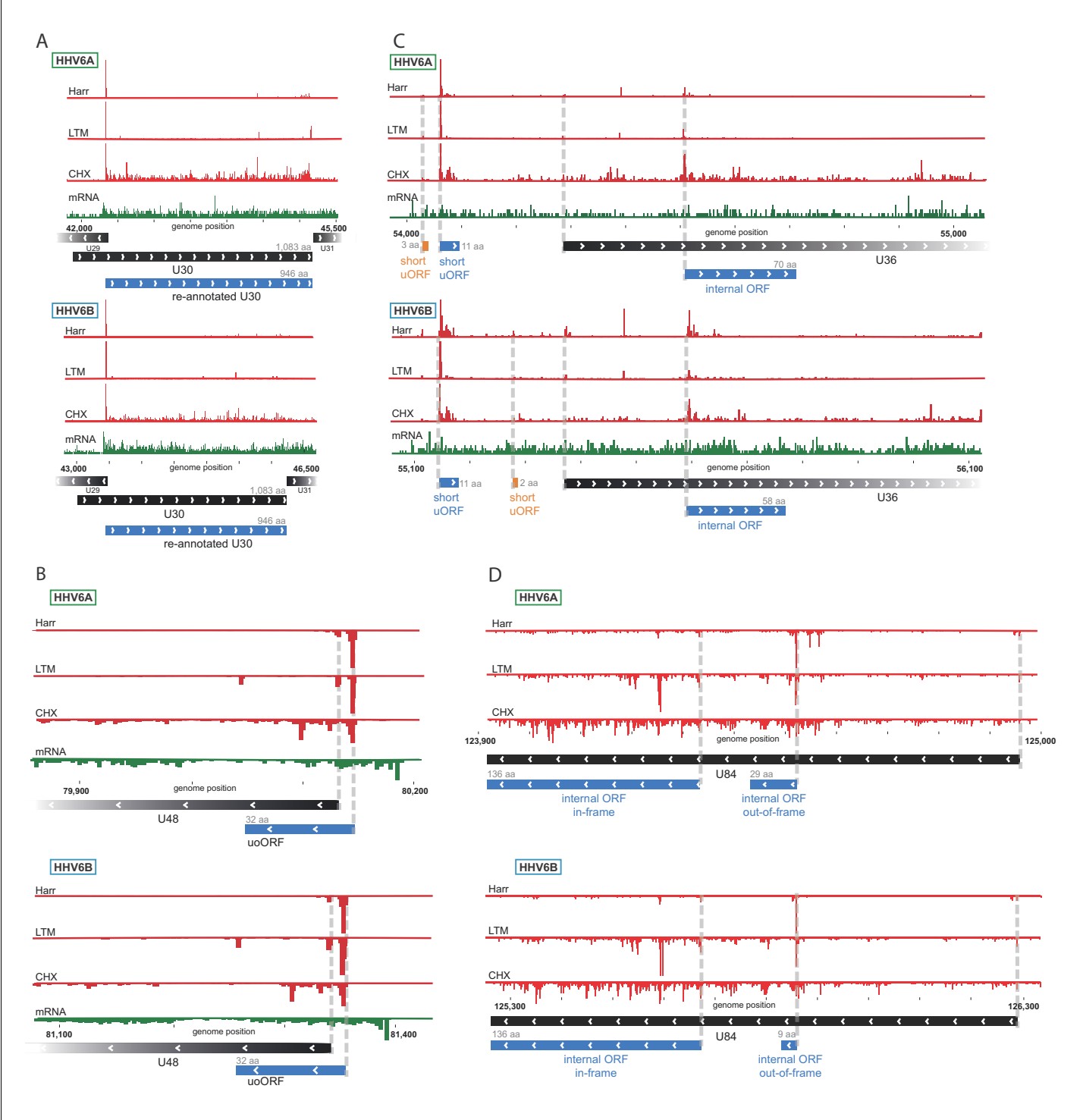

**Figure 2.** Ribo-Seq measurements reveal the architecture of viral coding regions. Examples of expression profiles of viral genes that contain novel ORFs conserved in HHV-6A and HHV-6B. Ribo-seq reads are presented in red and RNA-seq reads are presented in green. Canonical annotated ORFs are labeled by black rectangles, novel ORFs initiating at an AUG codon are labeled in blue, and novel ORFs initiating at a near-cognate start codon are labeled in orange. ORF sizes are written in gray. (A) U30 translation initiates at an AUG downstream of the annotated start codon. (B) A 32 amino acid (aa) upstream overlapping ORF (uoORF) is coded by the U48 transcript, initiates upstream of the U48 canonical ORF and partially overlaps it. (C) U36 locus contains two uORFs, as well as an out-of-frame iORF. (D) U84 locus contains an in-frame iORF which is a truncated version of U84, and a novel out-of-frame iORF.

example of a gene for which we found translation of two very short (<20 aa) uORFs from its 5'UTR (*Figure 2C*). In addition, we identified translation of an internal ORF (iORF), initiating out-of-frame, inside the coding region of U36 (*Figure 2C*), leading to translation of a novel ORF. In the U84 gene we observed two iORFs, one of them out-of-frame possibly regulating the downstream ORF, and another in-frame, starting at an AUG downstream of the U84 start-codon and ending in the same stop-codon, resulting in a truncated version of U84 (*Figure 2D*).

## RNA-seq analysis reveals pervasive splicing that is conserved between HHV-6A and HHV-6B

To systematically map the splice junctions of HHV-6A and HHV-6B, we used two independent splice-aware alignment tools, TopHat (*Trapnell et al., 2009*) and STAR (*Dobin et al., 2013*). We found an intricate set of splice junctions including dozens of novel splice junctions, which were overall positionally conserved between HHV-6A and HHV-6B (*Figure 3A* and *Figure 3—source data 1* and *2*). We were able to detect 24 out of 26 annotated HHV-6A splice junctions and all 24 annotated HHV-6B splice junctions. Furthermore, we identified 37 novel splice junctions in HHV-6A and 44 in HHV-6B (*Figure 3B* and *Figure 3—source data 1* and *2*). Some of the novel splice junctions identified in HHV-6A were recently reported in HHV-6B and are confirmed here for both viruses (U19, U83, and all splice forms of U79, *Greninger et al., 2018*). Interestingly, many of the novel splice junctions seem to belong to one long transcript composed of several short exons separated by long introns, spanning the U42-U57 locus (*Figure 3A*). In a few cases, novel splice junctions result in rean-notation of ORFs. For example, a splice junction between the HHV-6B U7 and U8 indicates that they are fused to one translation product, similar to the HHV-6A U7 (*Dominguez et al., 1999*; *Gompels et al., 1995*; *Gravel et al., 2013* and *Figure 3—supplement figure 1A*). Another splice junction in the HHV-6A U13 gene indicates that the U12 and U13 proteins share their N-terminal domain (*Figure 3—supplement figure 1B*). The same junction was also detected at lower levels in HHV-6B. The high relative abundance of reads that capture splice junctions suggests there is an extensive use of alternative splicing in these viruses.

## Previously unrecognized HHV-6 encoded long non-coding RNAs (lncRNAs)

By examining the RNA-seq data, we discovered three highly expressed novel transcripts, that lack both observed or potential long ORFs, suggesting that these are likely lncRNAs. These three lncRNAs are conserved between HHV-6A and HHV-6B, and they all contain efficiently translated short ORFs (*Figure 4A–C*). The short length of these ORFs implies that the RNAs themselves probably constitute functional elements. One lncRNA, designated here as lncRNA1, initiates within HHV-6 origin of replication (*Figure 4A*) and therefore resembles in synteny to an HCMV encoded lncRNA, RNA4.9 although it is much shorter (*Figure 4—figure supplement 1A*). This transcript is the most highly expressed polyadenylated RNA in both HHV-6A and HHV-6B (*Figure 4—figure supplement 2*), and its encoded short ORF, which contains the highest ribosome densities in the viral genomes (*Figure 4—source data 1*). The second lncRNA we identified, named here lncRNA2, is a spliced transcript that partially overlaps U18 (*Figure 4B*). The third lncRNA, designated lncRNA3, is transcribed between U77 and U79 (*Figure 4C*). This lncRNA has multiple possible isoforms generated by two alternative TSSs, two alternative polyadenylation sites, and alternative splicing. Initial inspection of the RNA-seq data suggested that the intron is not efficiently spliced (*Figure 4C*). However by synteny, this lncRNA is homologous to the HCMV encoded lncRNA5.0 and the Murine CMV encoded lncRNA7.2 (*Figure 4—figure supplement 1B*), shown to generate stable intronic RNAs which are not polyadenylated (*Kulesza and Shenk, 2006*; *Kulesza and Shenk, 2004*).

Since our RNA-seq libraries were based on poly-A selection and therefore non-polyadenylated RNA molecules are under-represented, we suspected similar intronic RNA products might be generated from lncRNA3. To explore this possibility, we quantified the number of reads that span the exon-intron junction relative to the number of intronic reads, and found that in both HHV-6A and HHV-6B they comprise less than 10% of what is expected from retained intron isoforms (*Figure 5A*). Therefore, these intronic reads do not seem to originate from intron retention and rather indicate that lncRNA3 also generates a stable non-polyadenylated intron. To further examine this possibility, we extracted RNA from cells infected with HHV-6A or HHV-6B and measured the abundance of

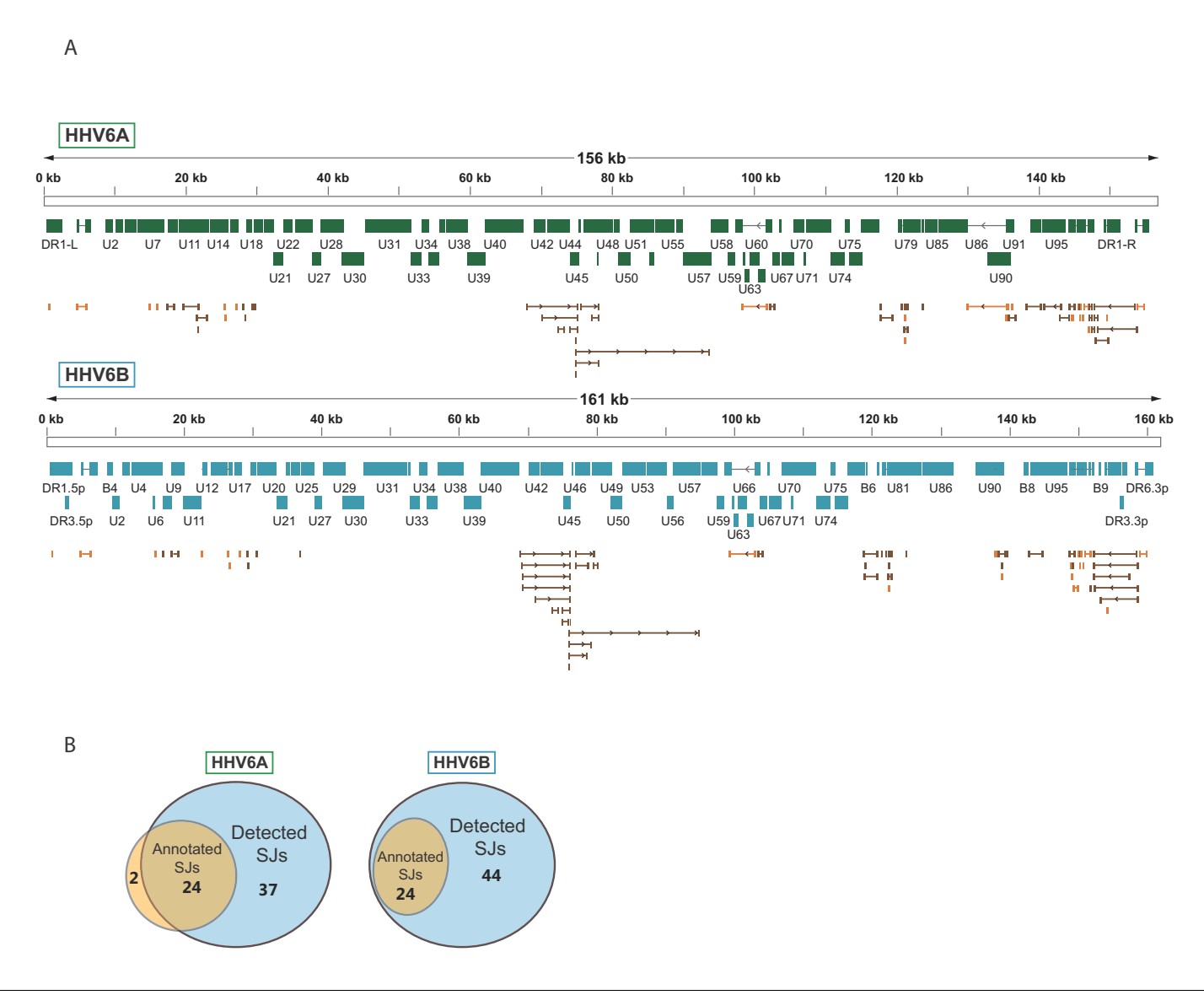

**Figure 3.** Splicing is abundant in HHV-6A and HHV-6B. (**A**) Splice junctions mapped using RNA-seq reads are shown throughout the genomes of HHV-6A and HHV-6B. Previously annotated splice junctions are marked in orange and novel splice junctions are marked in brown. (**B**) Diagrams displaying the numbers of previously annotated and detected splice junctions for HHV-6A and HHV-6B.

The online version of this article includes the following source data and figure supplement(s) for figure 3:

**Source data 1.** Splice junction annotation for HHV-6A. Scores from STAR and TopHat represent the number of reads covering the splice junction.
**Source data 2.** Splice junction annotation for HHV-6B. Scores from STAR and TopHat represent the number of reads covering the splice junction.
**Figure supplement 1.** Novel splice junctions result in reannotation of HHV-6 ORFs.

lncRNA3 intron in cDNA synthesized with random hexamers compared to cDNA synthesized with poly(dT) oligomers. Similar to the non-polyadenylated 18S ribosomal RNA, the intron RNA was detected at significantly higher levels in cDNA that was synthesized using random hexamers, while the polyadenylated lncRNA2 was more abundant or unchanged when poly(dT) oligomers were used in HHV-6A and HHV-6B, respectively (*Figure 5B*). We further quantified the abundance of the intronic RNA in of HHV-6B by deep sequencing total RNA without poly-A selection from infected cells. Based on these measurements the level of the intronic RNA of HHV-6B lncRNA3 is 100-fold higher than the spliced lncRNA3 (*Figure 5—figure supplement 1A*), making it the most abundant transcript in infected cells. Similarly, RNA-seq analysis of total RNA from HCMV infected cells

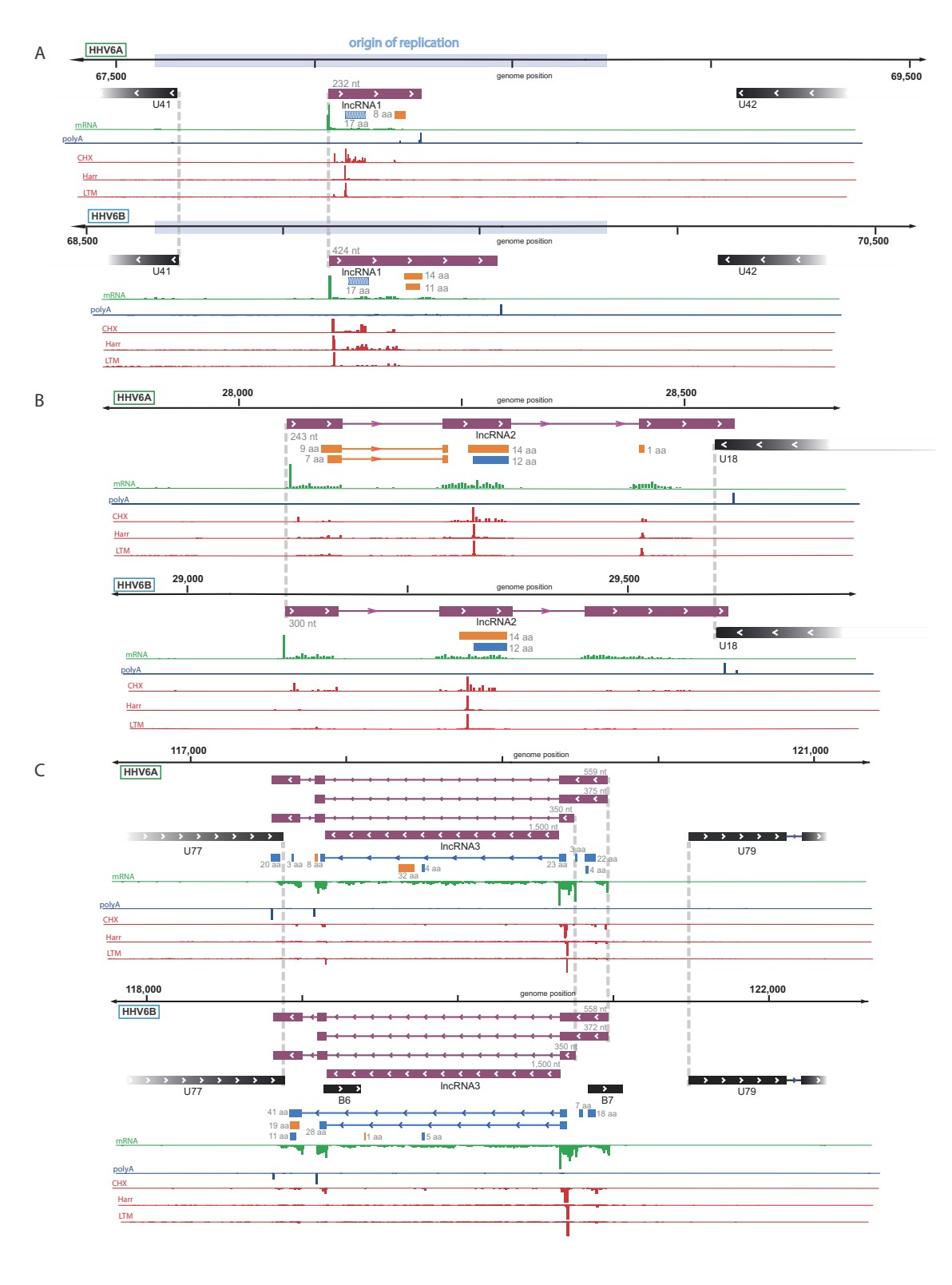

**Figure 4.** Identification of three highly abundant and conserved viral long non-coding RNAs (lncRNAs). Viral transcripts that appear to be lncRNAs are shown as purple rectangles. Reads from RNA-seq are presented in green and reads containing polyA are presented in blue. The ribosome profiling (CHX), Harringtonine (Harr) and lactimidomycin (LTM) profiles are presented in red. (**A**) A transcript initiating within the origin of replication. One

*Figure 4 continued on next page*

*Figure 4 continued*

putative ORF not detected by our predictions (see *Figure 6*) is shown as a striped blue rectangle. (B) A spliced transcript initiating between U17 and U18. (C) Three possible isoforms of a spliced transcript with alternative splicing, initiation and termination, as well as a putative stable intron.

The online version of this article includes the following source data and figure supplement(s) for figure 4:

**Source data 1.** RNA-seq and CHX Ribo-seq read density of previously annotated ORFs and novel lncRNAs for HHV-6A and HHV-6B.
**Figure supplement 1.** Conservation by synteny of newly discovered HHV-6 lncRNAs.
**Figure supplement 2.** RNA abundance of canonical ORFs and viral lncRNAs is conserved between HHV-6A and HHV-6B.

showed that the RNA5.0 intron is 10-fold higher than the spliced RNA5.0 (*Figure 5—figure supplement 1B*). Additionally, we validated the presence of the HHV-6B lncRNA3 intron by performing Northern blot analysis, confirming the presence of the RNA at the predicted size of ~1500 nt (*Figure 5—figure supplement 1C*).

Taken together, our results show that HHV-6 viruses express three highly abundant lncRNAs, as was shown in other herpesviruses (*Gatherer et al., 2011*; *Hutchinson and Tocci, 1986*; *Kulesza and Shenk, 2004*; *McDonough et al., 1985*; *Rawlinson and Barrell, 1993*), and that one of these lncRNAs, lncRNA3, generates a highly abundant stable non-polyadenylated intronic RNA that appears to be a conserved feature of betaherpesviruses.

## Systematic annotations of translated viral ORFs

To systematically define the full coding potential of HHV-6A and HHV-6B, we trained a support vector machine (SVM) model to identify translation initiation sites based on our Ribo-seq data sets, combining the actively translating ribosomes profile (CHX treatment), and initiation site enrichment (LTM and Harr) from cells infected with HHV-6 for 72 hr. The model was trained on a subset of the canonical viral ORFs that had high ribosome footprint coverage (see Materials and methods). Using the trained SVM model, we predicted hundreds of translation initiation sites in each virus (*Figure 6—source data 1* and *2*). In these sites, we found strong enrichment of translation initiation at the canonical AUG start codon, as well as weaker but still significant enrichment for the near-cognate start codons (*Figure 6A*). Of the near-cognate start codons, CUG was the most common, similar to what was found in other herpesviruses (*Arias et al., 2014*; *Stern-Ginossar et al., 2012*; *Whisnant et al., 2019*) and in human cells (*Fields et al., 2015*). Of the previously annotated ORFs, we identified translation in 69 out of 88 HHV-6A ORFs and 63 out of 103 HHV-6B ORFs. The ORFs missing from the prediction were either reannotated, or hardly translated under the conditions we used (*Figure 6—source data 3*). Since our detection is affected by the level of expression, it is likely these ORFs are expressed at low levels or translated under different conditions. In total, we identified 268 novel ORFs in HHV-6A and 216 novel ORFs in HHV-6B (*Figure 6B*). As expected, newly identified ORFs are shorter than the annotated ones (*Figure 6C*). Many of the novel ORFs we identified, were very short (<20 aa, 141 in HHV-6A and 111 in HHV-6B) and therefore are likely not functional at the polypeptide level. In addition, a large portion of the remaining ORFs are iORFs, translated within other ORFs (80 ORFs in HHV-6A and 67 in HHV-6B, *Figure 6B*). Due to the nature of the ribosome movement on the RNA during active translation, the ribosome protected fragments of coding sequences display a three-nucleotide periodicity, with enrichment for reads aligned to the first base of each codon. The newly identified ORFs displayed similar periodicity to the previously annotated ORFs, which was not seen in RNA-seq reads, further validating that these ORFs likely represent bona-fide translation products (*Figure 6D*).

## Pervasive use of alternative 5' transcript ends controls viral gene expression

Gene expression during lytic herpesvirus infection is regulated in a temporal cascade. In order to explore the temporal kinetics of HHV-6 ORFs we performed a time course experiment and created Ribo-seq and RNA-seq libraries from HHV-6B strain Z29 infected Molt-3 cells at 5, 24 and 72 hr post infection (hpi). For these experiments, we chose to focus on HHV-6B as it is more common and clinically relevant (*Braun et al., 1997*; *Clark, 2016*). The data in this experiment was highly correlated with our single time point experiment (Pearson's R on log transformed data is 0.98 for RNA-seq 0.97 and for Ribo-seq, *Figure 7—figure supplement 1*).

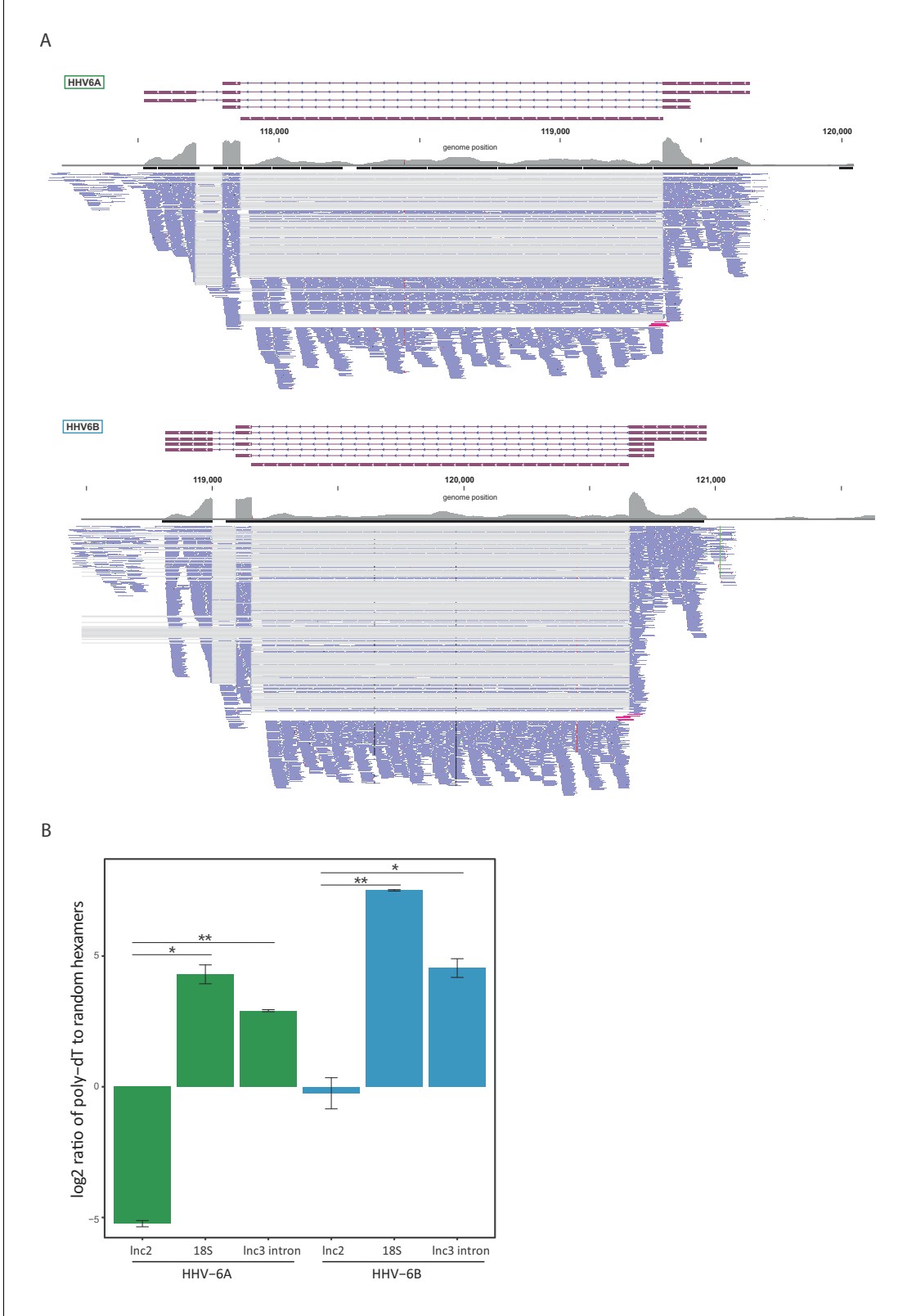

**Figure 5.** lncRNA3 generates a stable non poly adenylated intron. (**A**) RNA-seq reads aligned to the negative strand of lncRNA3 locus in both HHV-6A and HHV-6B are presented. Thin gray lines represent spliced reads, blue lines represent reads aligned to either the exons or intron, pink lines represent reads that span the first exon intron junction. In regions with very high coverage (>100 reads per 50 nt region) reads were downsampled so that maximum 100 reads per region are displayed. Gray bars represent the total reads coverage without omissions. (**B**) RT-qPCR measurements of the HHV-
*Figure 5 continued on next page*

*Figure 5 continued*

6A and HHV-6B lncRNA3 intron RNA. Values were normalized to the HHV-6 U21 gene. cDNA was prepared with either oligo-dT or random hexamers primers and the ratio of these measurements is presented. Error bars represent standard error of biological duplicates. P-values were calculated using Student's t-test. * p-value<0.05 and ** p-value<0.01.

The online version of this article includes the following figure supplement(s) for figure 5:

**Figure supplement 1.** Abundance of lncRNA stable intron in total RNA.

Hierarchical clustering of viral coding regions by footprint densities along infection (a measure of the relative translation rates) revealed several distinct temporal expression patterns (*Figure 7A* and see *Figure 7—source data 1* for read numbers). These temporal profiles largely agree with previously published kinetic classifications (*Tsao et al., 2009*; *Yamanishi et al., 2013*). Cluster one contains ORFs whose expression is relatively high at 5hpi compared to 24 and 72hpi, and this cluster includes most of the genes classified as immediate-early (IE, U79, U90 and U95). Another gene, U85, a glycoprotein previously classified as IE (*Tsao et al., 2009*), was not efficiently translated at 5hpi and was assigned to cluster 2. Cluster two contains genes that are most highly expressed at 24hpi and is enriched in early genes. Clusters 3 and 4 contain genes that are mostly expressed at 72hpi and are both enriched in late genes; however, cluster four is composed of genes that are expressed almost exclusively at 72hpi. While most of the previously annotated late genes were assigned to these clusters, the DNA helicase/primase U43 and the large tegument protein U31 were previously annotated as late genes, but are shown here to reach peak translation at the 24hpi timepoint.

We previously demonstrated that pervasive use of alternative 5' ends in HCMV transcripts is critical for the tight temporal regulation of viral gene expression and production of alternate protein products (*Stern-Ginossar et al., 2012*). We observed similar phenomena in the temporal regulation of several HHV-6B genes. For example, the U53 gene contains newly identified iORFs, one of which initiates at an AUG, and is an orthologue of the annotated HHV-6A U53.5 ORF (*Figure 7B*). Relative to the main U53 ORF, these iORFs are translated more efficiently at 72hpi than at 24hpi. This could be explained by a temporal shift in the relative frequency of initiation at two TSSs, one of which is upstream of the U53 start codon from which the main U53 can be translated, and another downstream of the U53 start codon allowing translation of the iORFs but not of the main U53 ORF. Notably, we found the same pattern in the HCMV homolog, UL80 (*Stern-Ginossar et al., 2012*; *Figure 7B*). A similar form of regulation is seen in the HHV-6B locus coding for the U81 and U82 ORFs in which we found two TSSs. One TSS is immediately upstream of U81 creating an RNA that is mainly expressed at 24hpi, facilitating the translation of U81. At 72hpi a second TSS is also present, giving rise to translation of U82 (*Figure 7C*). Temporal regulation of 5' ends was also found for the HHV-6B U51 and its uoORF, which is also conserved in its HCMV homolog UL78 (*Stern-Ginossar et al., 2012*; *Figure 7—figure supplement 2*).

## uORFs are enriched in betaherpesvirus late genes

Among the newly identified ORFs many are iORFs and uORFs. Since the high abundance of these ORFs may be associated with changes in translation regulation, we examined whether these translation events are enriched in specific kinetic classes. Each uORF and iORF was assigned to a canonical transcript; iORFs were assigned to the canonical ORF in which they reside, and uORFs were assigned to a canonical ORF if they were located upstream of its translation initiation (*Figure 7—source data 2*). For both HHV-6A and HHV-6B we found an enrichment of uORFs in the 5'UTRs of late genes compared to earlier kinetic classes (*Figure 7D*, p-value < 0.01, proportion test). In contrast, there was no enrichment for the presence of iORFs in any kinetic class (*Figure 7E*, p-value > 0.3), negating the option that the enrichment we found for uORFs is due to a bias in our approach or to a general increase in our ability to capture translation initiation. We further extended this analysis to HCMV ORFs and found that uORFs but not iORFs are enriched in 5'UTRs of HCMV genes that are expressed with late kinetics, similar to what we see in HHV-6 (*Figure 7D and E*). Since the ability to capture uORFs translation might be affected by expression levels, which can skew our analysis, we checked the correlation between RNA expression and the number of predicted uORFs. We identified a positive correlation (Pearson's R = 0.34) for HHV-6B ORFs, but not for HHV-6A or HCMV ORFs (R = 0.04 and R = 0.03 respectively), suggesting expression levels probably do not solely

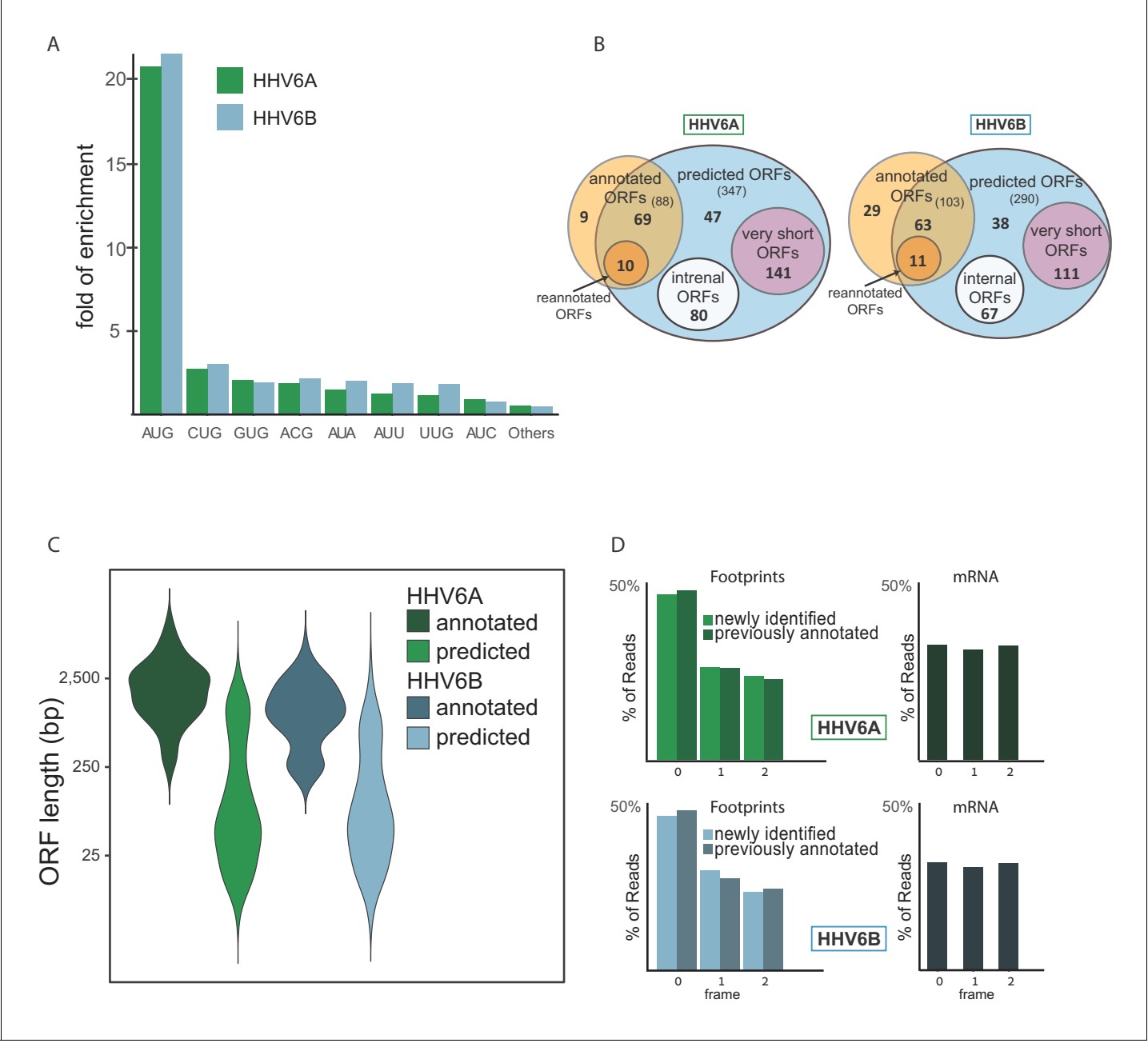

**Figure 6.** Identification of hundreds of novel HHV-6 ORFs. (**A**) Fold enrichment of AUG and near-cognate codons at predicted sites of translation initiation compared to their genomic distribution. (**B**) Venn diagrams summarizing the HHV-6 translated ORFs. (**C**) Size distribution of previously annotated ORFs (dark) and of newly identified ORFs (bright). (**D**) Position of the ribosome footprint reads relative to the translated reading frame showing enrichment of the first position in the annotated ORFs (dark) as well as in the newly identified ones (bright). The mRNA reads were used as control and do not show enrichment to any frame.

The online version of this article includes the following source data for figure 6:

**Source data 1.** SVM predicted ORFs in HHV-6A.
**Source data 2.** SVM predicted ORFs in HHV-6B.
**Source data 3.** Previously annotated ORFs not included in the final predictions.

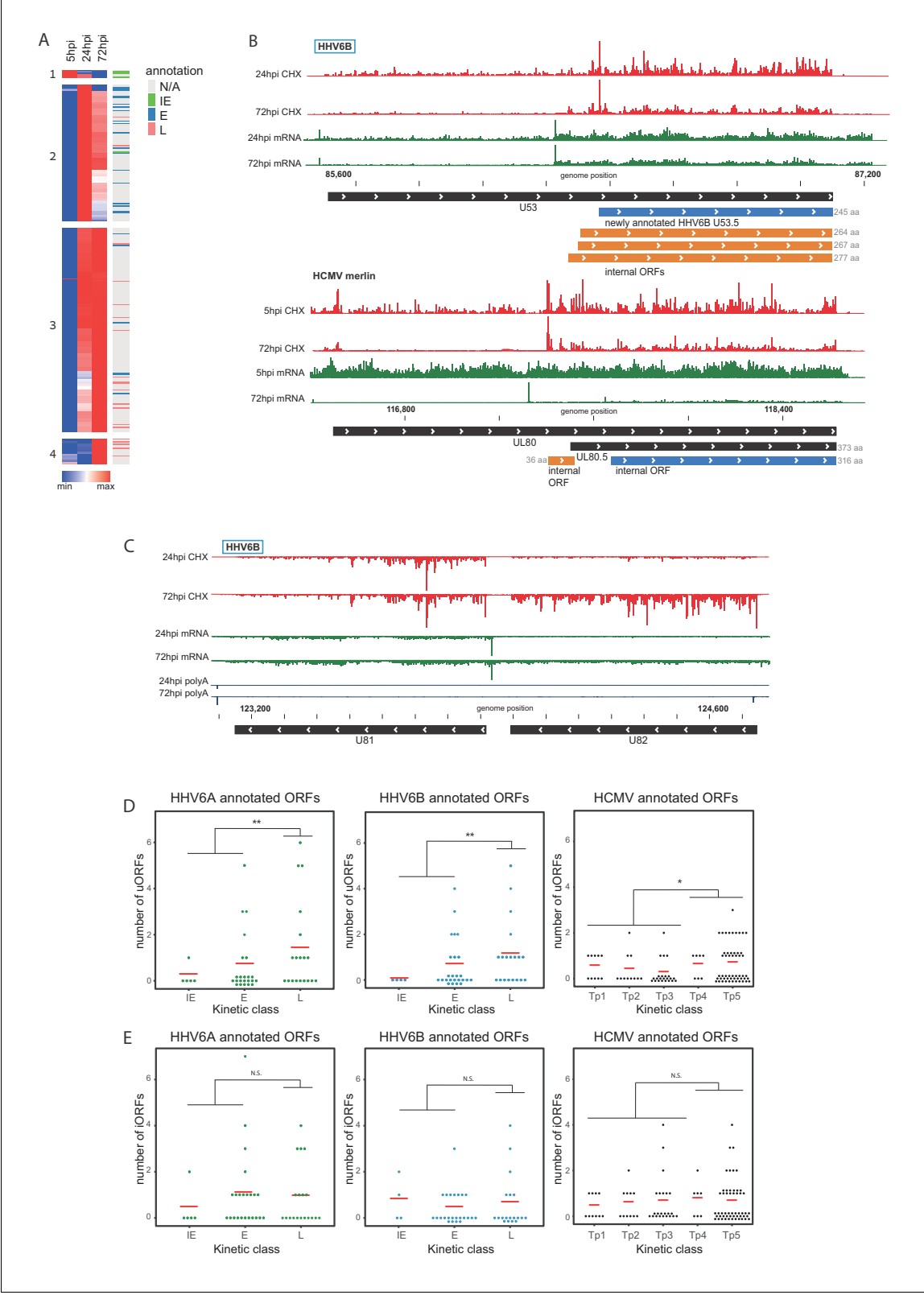

**Figure 7.** Temporal regulation of viral gene expression is driven by pervasive use of alternative 5' ends. (**A**) Heatmap of ribosome occupancy of HHV-6B ORFs clustered by relative expression levels at 5, 24 and 72hpi. Previously annotated kinetic class were labeled on the right as immediate early (IE, green), early (E, blue), late (L, pink), or unknown (N/A, gray). The cluster number appears on the left. (**B and C**) The ribosome occupancy (red) and mRNA profiles (green) are shown (**B**) around U53 loci at different hours post infection (marked on the left) and around its HCMV homolog, UL80 (**C**) and

*Figure 7 continued on next page*

*Figure 7 continued*

around U81 and U82 loci. (**D and E**) Dot plots showing the number of uORFs (**D**) and iORFs (**E**) of each canonical viral ORF with annotated kinetic class for HHV-6A, HHV-6B and HCMV. P-value was calculated using proportion test. * for p-value<0.05, ** for p-value<0.01 and N.S for non-significant.

The online version of this article includes the following source data and figure supplement(s) for figure 7:

**Source data 1.** CHX Ribo-seq density and kinetic clusters of SVM predicted HHV-6B ORFs.
**Source data 2.** Internal and upstream ORFs of previously annotated HHV-6A and HHV-6B ORFs and their HCMV homologs.
**Figure supplement 1.** RNA abundance and ribosome footprint coverage correlate well between replicates.
**Figure supplement 2.** Conserved temporal regulation of translation from uoORF.
**Figure supplement 3.** Number of uORFs as a function of RNA abundance.
**Figure supplement 4.** Enrichment of non-AUG start codons at late time points post infection.

explain the enrichment we see for uORFs in late genes (*Figure 7—figure supplement 3*). Altogether, these results suggest a potential mechanism for translation regulation of late viral genes, utilizing uORFs, which is conserved among betaherpesviruses. Interestingly, we also observed an increased proportional use of non-canonical start codons late in infection (*Figure 7—figure supplement 4*), further supporting the possibility that a change in translation regulation might occur at late time points post infection.

## The presence of iORFs and uORFs is conserved among betaherpesvirus genes

Using our comprehensive transcriptome and translatome data we uncovered hundreds of novel ORFs in HHV-6A and in HHV-6B. We next examined whether the presence of these ORFs is conserved between these two HHV-6 species. We found that the number of iORFs and uORFs in HHV-6A and HHV-6B homolog ORFs are well correlated, indicating a high level of conservation of these translation events between these two viruses (p<$10^{-15}$ for uORFs and p<$10^{-10}$ for iORFs, *Figure 8A*). Several homolog ORFs have multiple conserved iORFs and/or uORFs (*Figure 8B* and *Figure 8—figure supplement 1*). We also found some features that are conserved in HCMV. In five iORF-containing HHV-6 genes and in four uORF-containing HHV-6 genes, the HCMV homologs also contained similar iORFs or uORFs (*Figure 8—figure supplement 2*). One of the HHV-6/HCMV homolog ORF pairs containing a conserved uORF is U51 and its HCMV homolog UL78 (*Figure 8C*), which interestingly also show conserved kinetics along infection suggesting a potential regulatory mechanism conserved between these viruses (*Figure 7—figure supplement 2*). Altogether, the conserved presence of several uORFs and iORFs suggests that their occurrence is not random, and it is likely that these represent a functional module that plays a role in regulating herpesvirus protein expression.

## Discussion

Decoding the transcriptional and translational landscape of any virus is a fundamental step in studying its biology and pathogenesis. For many herpesvirus genomes, traditional annotations have relied on the identification of canonical translational start codons and arbitrary size restriction to define viral open reading frames (ORFs). Laborious follow-up molecular work revealed the transcriptional architecture of individual genomic loci, but for most HHV-6 genes annotations are still based on these initial in-silico ORF predictions. In recent years, major advances in high-throughput sequencing approaches have revealed that the transcriptome and translatome of herpesviruses are extremely complex, encompassing large numbers of overlapping transcripts, extensive splicing and many non-canonical translation products (*Arias et al., 2014*; *Balázs et al., 2017*; *Bencun et al., 2018*; *Depledge et al., 2019*; *Gatherer et al., 2011*; *Kara et al., 2019*; *O'Grady et al., 2019*; *O'Grady et al., 2016*; *Tombácz et al., 2017*; *Whisnant et al., 2019*). Our own work in which we employed ribosome profiling and systematic transcript analysis to decipher HCMV genome complexity revealed a rich collection of coding sequences that include many viral short ORFs (sORFs), uORFs and alternative translation products that generate extensions or truncations of canonical proteins (*Stern-Ginossar et al., 2012*).

Here, using RNA-Seq and ribosome profiling measurements along HHV-6 infection, we provide a comprehensive map of HHV-6A and HHV-6B coding elements over the lytic life cycle. In agreement

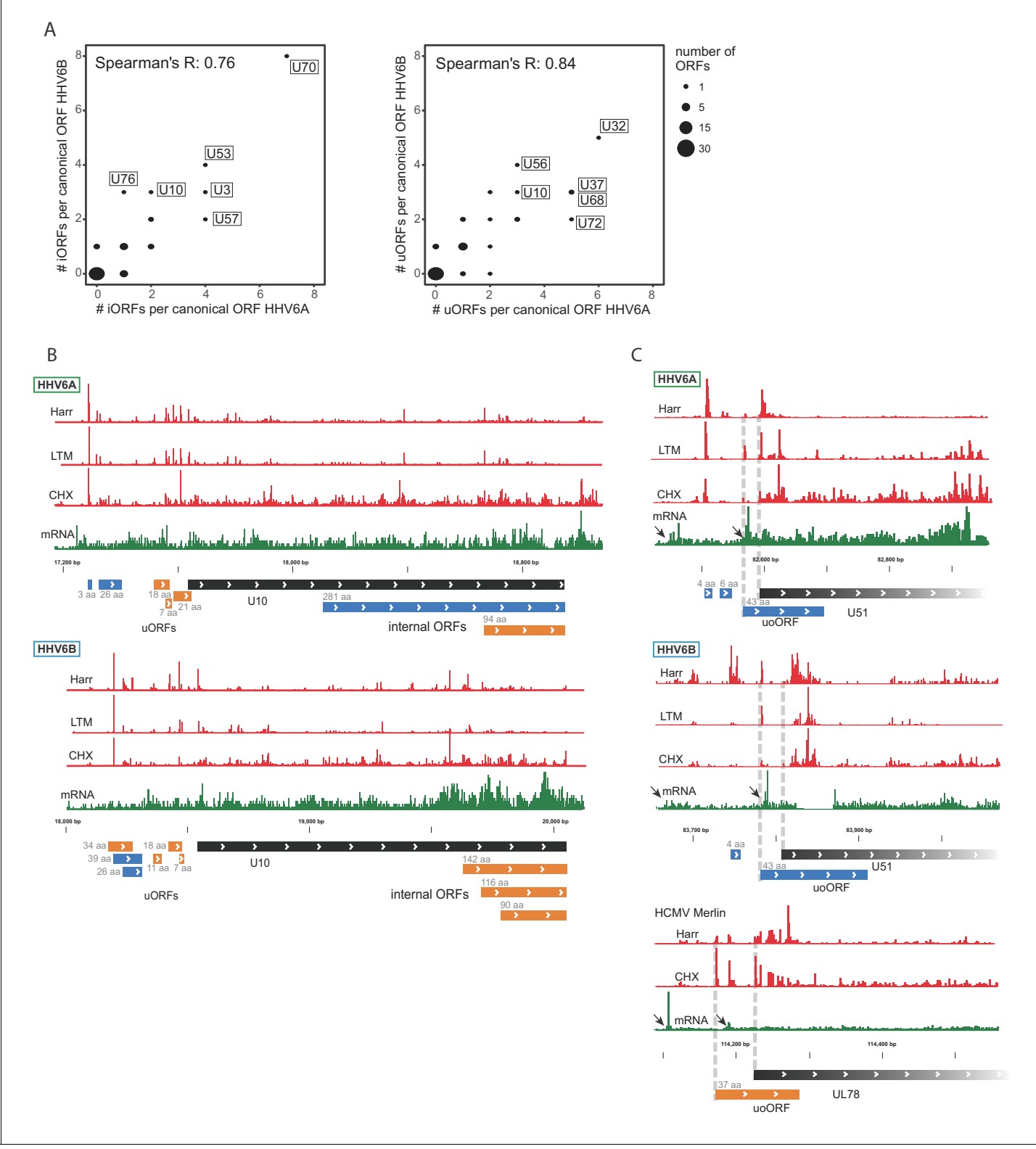

**Figure 8.** Numerous iORFs and uORFs are conserved between betaherpesviruses. (**A**) Correlation between the number of iORFs and uORFs of canonical ORFs in HHV-6A and HHV-6B (55 shared canonical ORFs in total). Dot size indicates the number of canonical ORFs with the indicated number of iORFs or uORFs in the two viruses. (**B–C**) Selected examples of novel internal or upstream initiation events that are conserved between HHV-6A and HHV-6B. Shown in black rectangles are canonical ORFs, in blue are novel ORFs initiating at an AUG codon, and in orange are novel ORFs initiating at a near-cognate start codon. ORF sizes are written in gray. The ribosome occupancy profiles are shown in red and the mRNA profile is shown in green (**B**)

*Figure 8 continued on next page*

*Figure 8 continued*

at U10 locus for both HHV-6A and HHV-6B and (C) at the U51 locus in HHV-6A and HHV-6B and its HCMV homolog U78. The gap in RNA reads in HHV-6B U51 is due to a base insertion relative to the reference, preventing read alignment to the region.

The online version of this article includes the following figure supplement(s) for figure 8:

**Figure supplement 1.** Viral loci with conserved presence of multiple uORFs and iORFs.
**Figure supplement 2.** Synteny conservation of uORFs and iORFs between HHV-6 and HCMV.

with the complexity of other herpesviruses that have been analyzed using ribosome profiling approaches, we identified 268 and 216 novel viral ORFs that are expressed during HHV-6A and HHV-6B lytic infection, respectively. Furthermore, our transcriptome analyses enabled mapping of the full landscape of HHV-6 splice junctions and the identification of three virally encoded lncRNAs. Our data further show that in similarity to our findings in HCMV (*Stern-Ginossar et al., 2012*), the pervasive use alternative of 5' ends plays a major role in HHV-6 genomes in production of distinct polypeptides from single genomic loci. Like alternative splicing, this mechanism can expand protein diversity and contribute to virus complexity by allowing multiple distinct polypeptides to be generated from a single genomic locus. Overall, our revised experimental annotations will facilitate functional studies on HHV-6 ORFs and transcripts as well as their regulation.

This wealth of novel elements requires a more precise dissection of the components that are likely to be functional. The issue of functional relevance still represents a major challenge in these systematic experimental annotations. Our analysis of three betaherpesviruses allowed us, for the first time, to highlight some conserved features that may point towards functional importance. A large portion of the novel translated ORFs we identified are uORFs. uORFs are widely recognized as *cis*-regulatory elements and their presence generally correlates with reduced translation of the primary ORF, but there are instances in which they associate with increased translation (*Young and Wek, 2016*). Despite their pervasiveness, only a few viral uORFs have been studied in detail (*Geballe et al., 1986*; *Kronstad et al., 2013*). We show that genes that contain uORFs and the number of uORFs are largely conserved between HHV-6A and HHV-6B. In addition, we reveal that both in HHV-6 and in HCMV, uORFs appear to be especially abundant in late viral genes. The surplus of uORFs and their preferred use specifically at late time points of infection indicate that they may have a functional role in controlling viral gene expression, probably when cellular stress pathways are engaged. The overall high representation of sORFs, iORFs and uORFs in the viral genome, particularly at late stages of infection, is probably driven by several mechanisms; the first is extensive use of alternative transcription initiation, which is highly prevalent at late time point of infection with herpesviruses, and allows the translation of multiple translation products from the same viral loci (*Balázs et al., 2018*; *Parida et al., 2019*; *Stern-Ginossar et al., 2012*; *Whisnant et al., 2019*); the second mechanism may be related to changes in translation permissiveness which can explain the increase in translation initiation from near canonical start codons. Notably, our expression data along infection likely underestimate the true changes in viral protein production as experimental HHV-6 infection inherently creates a mixed population of cells infected at different times (see Materials and methods).

We identified three conserved HHV-6 encoded lncRNAs, signifying lncRNAs are probably a shared feature of all herpesviruses (*Tycowski et al., 2015*). lncRNAs are still an enigmatic group of RNA molecules that do not form a well-defined class of genes, and mechanistically most lncRNAs, including viral lncRNAs, remain poorly characterized. Unlike mammalian lncRNAs, that as a group are significantly less abundant than canonical mRNAs (*Mukherjee et al., 2017*), in herpesviruses lncRNAs represent the most abundant group of viral transcripts (*Gatherer et al., 2011*; *Tycowski et al., 2015*). These high expression levels allude to essential roles for virally encoded lncRNAs during infection. The three HHV-6 encoded lncRNAs we identified are highly expressed but present distinct features; lncRNA1 is relatively short (232 bp in HHV-6A and 424 bp in HHV-6B) and unspliced, lncRNA2 is composed of three exons that are efficiently spliced, and lncRNA3 represents a complex locus with two different TSSs and polyadenylation sites, three alternatively spliced exons and a stable intron.

Interestingly, by synteny lncRNA1 and lncRNA3 seem related to HCMV encoded lncRNAs. lncRNA1 resembles the HCMV RNA4.9 as both are transcribed from the viral origin of replication at the same orientation. This similarity implies a possible conserved role of lncRNA transcription in

betaherpesviruses origin of replication although they are very different in length (RNA4.9 is 4.9 kb long). lncRNA3 is an orthologue of HCMV encoded RNA5.0 both in synteny and in the production of a stable intron. RNA5.0 has previously been shown to generate a stable intron that is not required for HCMV replication in fibroblasts (*Kulesza and Shenk, 2004*). A murine cytomegalovirus 7.2 kb ortholog of RNA5.0 was identified which also generates a stable intronic RNA. Mutant MCMV viruses lacking this stable intron RNA replicated normally in culture, but exhibited a defect in establishing a persistent infection in vivo (*Kulesza and Shenk, 2006*). Our results indicate that the production of a stable intronic RNA from this locus is a conserved feature of betaherpesviruses, implying a central function. Importantly, the notion that this non-coding region is conserved in betaherpesviruses and therefore likely represents a functional component was already specified 15 years ago (*Dolan et al., 2004*). The strong expression of the intronic RNA, which is 100-fold higher compared to the spliced RNA, and its conservation in beta herpesviruses points that this intronic RNA is probably the main functional element in this locus. This may also explain the apparent complexity of the locus, if the intronic RNA mediates the function, there will be no selection to maintain a specific RNA isoform or a specific transcription start site as long as the intron is generated, allowing multiple isoforms to arise from the same locus.

The high abundance of this RNA together with its conservation make the molecular and functional characterization of these viral intronic RNAs highly interesting. There is little known about the mechanisms by which stable intronic RNAs may operate (*Osman et al., 2016*) but one possibility is that these RNAs sequester spliceosomes or specific splicing components that cause changes in the cellular splicing activity. Interestingly, a small non-coding RNA (sncRNA-U77) that is mapped to the intron of lncRNA3 was shown to be expressed by HHV-6A (*Nukui et al., 2015*). It is therefore possible that the stable intron is further processed to create additional functional elements. Furthermore, a recent study showed that TSA-mediated HHV-6A transactivation results in increased transcription from a region overlapping the lncRNA3 locus (*Prusty et al., 2018a*), implying lncRNA3 may be involved in HHV-6A reactivation.

In conclusion, we provide a comprehensive annotation of HHV-6 transcripts and ORFs and highlight conserved translation patterns and non-coding RNAs that may have central shared functions in all betaherpesviruses.

# Materials and methods

## Key resources table

| Reagent type (species) or resource | Designation | Source or reference | Identifiers | Additional information |
|---|---|---|---|---|
| Strain, strain background (HHV-6A) | GS | NIH AIDS | | |
| Strain, strain background (HHV-6B) | Z29 | NIH AIDS | | |
| Cell line (*Homo-sapiens*) | HSB-2 | NIH AIDS, Electro-Nucleonics, Inc (*Barre-Sinoussi et al., 1983*) | | |
| Cell line (*Homo-sapiens*) | Molt-3 | NIH AIDS | ATCC CRL1552 | |
| Sequence-based reagent | lncRNA3-6A F | This paper | qPCR primers | AAAAGGACAAGAG CAGCCGC |
| Sequence-based reagent | lncRNA3-6A R | This paper | qPCR primers | ACTCGTATCACCTAC CTCTCTCTAC |
| Sequence-based reagent | lncRNA3-6A F | This paper | qPCR primers | GGTATCGGGGTAAG AATAAGATGACG |
| Sequence-based reagent | lncRNA3-6A R | This paper | qPCR primers | AAAAGGACAAGAGC AGCCGC |

*Continued on next page*

*Continued*

| Reagent type (species) or resource | Designation | Source or reference | Identifiers | Additional information |
|---|---|---|---|---|
| Sequence-based reagent | lncRNA2-6B F | This paper | qPCR primers | CAAAACGGTCTCAC TGCTCC |
| Sequence-based reagent | lncRNA2-6B R | This paper | qPCR primers | TCTATAAAGTGCC GTGAGTGC |
| Sequence-based reagent | lncRNA2-6A F | This paper | qPCR primers | CGACAAAACAAAAT AGTCCCACT |
| Sequence-based reagent | lncRNA2-6A R | This paper | qPCR primers | ATGGAAAAGGT GGTCGTGGA |
| Sequence-based reagent | U21-6B F | This paper | qPCR primers | CCGCACCCATGA ACATAAGG |
| Sequence-based reagent | U21-6B R | This paper | qPCR primers | ATGATGTGACGTG GGGACTT |
| Sequence-based reagent | U21-6A F | This paper | qPCR primers | CCAGCCACCTAGA GAACGAA |
| Sequence-based reagent | U21-6A R | This paper | qPCR primers | TTGGGCTGAACTC TCGACAT |
| Sequence-based reagent | 18 S F | This paper | qPCR primers | CTCAACACGGGAA ACCTCAC |
| Sequence-based reagent | 18 S R | This paper | qPCR primers | CGCTCCACCAACTA AGAACG |
| Sequence-based reagent | probe 1 F | This paper | Northern blot probe template primers | GTAAGATTTAACCT ATTTTGCAT |
| Sequence-based reagent | probe 1 R | This paper | Northern blot probe template primers | TAATACGACTCACTA TAGGGTGA TGACAATATAGAAGATGG |
| Sequence-based reagent | probe 2 F | This paper | Northern blot probe template primers | GAAAAGTCATCAGAAAA GTCATCAGAA |
| Sequence-based reagent | probe 2 R | This paper | Northern blot probe template primers | TAATACGACTCACTATAGGG TCA ACTGTTTTGTGCCCAAC |
| Sequence-based reagent | probe 3 F | This paper | Northern blot probe template primers | TATTTAGTTCACATTA TAAGGACCT |
| Sequence-based reagent | probe 3 R | This paper | Northern blot probe template primers | TAATACGACTCACT ATAGGGCT GCAAAAACAAATGA AAGTCT |
| Software, algorithm | Bowtie v1.1.2 | (*Langmead et al., 2009*) | | |
| Software, algorithm | Morpheus | https://software. broadinstitute.org/ morpheus | | |
| Software, algorithm | TopHat v2.1.1 | (*Kim et al., 2013*; *Trapnell et al., 2009*) | | |
| Software, algorithm | STAR v2.5.3a | (*Dobin et al., 2013*) | | |
| Software, algorithm | R 3.6.0 | (*R Development Core Team, 2019*; *Wickham, 2016*) | | |

## Cell lines and virus strains

HSB-2 Cells from Electro-Nucleonics, Inc (*Barre-Sinoussi et al., 1983*) and Molt-3 cells (ATCC CRL1552) were maintained at 37°C in 5% (vol/vol) $CO_2$, in RPMI 1650 medium (Biological Industries) supplemented with 10% heat-inactivated fetal bovine serum (Life Technologies), 2 mM L-glutamine

(Biological Industries), 1 mM sodium pyruvate, 0.1 mg/mL streptomycin and 100 U/mL penicillin (Biological Industries). Cell line identity was authenticated by confirming their morphology and growth rate corresponded to source description. All cells were tested and found negative for Mycoplasma.

HHV-6A strain GS and HHV-6B strain Z29 were maintained in HSB2 and Molt-3 cells, respectively. For viral propagation, infected cells were added to uninfected cells at a ratio of 1:10 every 3 or 4 days. All viruses and cell lines were obtained from the NIH AIDS reagent program, Division of AIDS, NIAID, NIH.

## Preparation of ribosome profiling and RNA sequencing samples

Samples were prepared by co-incubating about 7 million cells of either HSB-2 or Molt-3 at a density of 1–1.5 M cells per mL with cells infected with HHV-6A and HHV-6B, respectively, at a 1:5 ratio, for 72 hr.

For RNA-seq, cells were harvested with Tri-Reagent (Sigma-Aldrich), total RNA was extracted, and poly-A selection was performed using Dynabeads mRNA DIRECT Purification Kit (Invitrogen). For Ribo-seq libraries, cells were treated with either 50 µM lactimidomycin (LTM) for 30 min or 2 µg/mL Harringtonine (Harr) for 5 min, for translation initiation libraries (LTM and Harr libraries correspondingly), or left untreated for the translation elongation libraries (cycloheximide [CHX] library). All three samples were subsequently treated with 100 µg/mL CHX for 1 min. Cells were placed on ice immediately after treatment, centrifuged and washed twice with PBS containing 100 µg/mL CHX. Subsequent lysis, Ribo-seq and RNA-seq library generation were performed as previously described (*Ingolia et al., 2011*).

For HHV-6B infection kinetics, virus containing supernatant was collected from Molt-3 cells infected for four days. Six samples of 250,000 Molt-3 cells were incubated in 50 µL of the viral supernatant each, for 30 min at 4°C and then for 45 min at 37°C. After infection, the cells were incubated in RPMI at a cell density of 1 million per 1.5 mL. Cells were harvested at 5, 24 and 72 hr post infection, and CHX and RNA-seq libraries were generated as described above. For total RNA sequencing without poly-A selection, total RNA from HHV-6B infected Molt-3 cells and HCMV infected HFFs at 72hpi was extracted as described and libraries were created using SENSE Total RNA-Seq Library Prep Kit (Lexogen). Prepared libraries were sequenced on the illumina NextSeq 500 with at least 61nt single-end reads.

## Northern blot analysis

Total RNA was extracted from Molt-3 cells infected with HHV-6B for 72 hr as described above. Northern blot was performed using the NorthernMax kit (Ambion), mostly according to manufacturer's instructions. In short, 5 µg of total RNA was run on a 1% denaturing agarose gel. After RNA transfer from the agarose gel onto the BrightStar nylon membrane (Ambion), it was crosslinked to the membrane using UV radiation. Both pre-hybridization and hybridization steps were performed over-night at 68°C and a mix of three different RNA probes (0.1 nM each) was used to hybridize with the target RNA. Subsequently, the membrane was washed and then incubated in blocking buffer (Odyssey Blocking Buffer PBS (Licor) containing 0.5% SDS) for 30 min at room temperature. Next, the membrane was incubated with Alexa Fluor 647 Streptavidin (Biolegend) in blocking buffer in a dilution of 1:10 000. The membrane was washed with PBST and analyzed using Odyssey CLx (Licor). For size estimation of the transcript of interest, the 28S and 18S rRNA bands were used.

For probe generation, the lncRNA3 intron sequence was amplified from cDNA using PCR reactions with following primers:

probe 1 F GTAAGATTTAACCTATTTTGCAT
probe 1 R TAATACGACTCACTATAGGGTGATGACAATATAGAAGATGG
probe 2 F GAAAAGTCATCAGAAAAGTCATCAGAA
probe 2 R TAATACGACTCACTATAGGG TCAACTGTTTTGTGCCCAAC
probe 3 F TATTTAGTTCACATTATAAGGACCT
probe 3 R TAATACGACTCACTATAGGGCTGCAAAAACAAATGAAAGTCT

In vitro transcription was performed using the MegaScript Kit (Ambion) according to manufacturer's instructions.

## Sequence alignment, normalization, metagene analysis and clustering and visualization

Sequencing reads were aligned as previously described (*Tirosh et al., 2015*). Briefly, linker and poly-A sequences were removed and the remaining reads were aligned to the human and the viral reference genomes (HHV-6A KC465951.1, HHV-6B AF157706.1) using Bowtie v1.1.2 (*Langmead et al., 2009*) with maximum two mismatches per read. Reads that were not aligned to the genome were aligned to the transcriptome, taking into account all the new identified splice junctions. Reads aligned to multiple locations were discarded, therefore, genomic repeat regions were not included in the analysis. Sequencing data was visualized using IGV integrative genomics viewer (*Robinson et al., 2011*).

For the metagene analysis only genes with more than 100 reads were used. Each gene was normalized to its maximum signal and each position was normalized to the number of genes contributing to the position.

For the time-course clustering, footprints counts of one sample from each time point of HHV-6B infected cells were normalized to units of reads per million (RPM) in order to normalize for sequencing depth. To avoid noise arising from low viral gene expression at 5hpi, ORFs with less than six reads at this time point were considered to have zero reads. Morpheus (https://software.broadinstitute.org/morpheus) was used to perform hierarchical gene clustering with one minus Pearson correlation as metric and complete linkage method.

For comparing transcript expression level, mRNA and footprint counts were normalized to units of reads per kilobase per million (RPKM) in order to normalize for gene length and for sequencing depth.

Single nucleotide mutations in RNA-seq were identified (*Mizrahi et al., 2018*) and positions with at least 10 reads that had a different base than the reference in 95% or more of the reads are listed in *Supplementary files 1* and *2*. Lists of deletions and insertions that scored 20 or above in the TopHat output are also in *Supplementary files 1* and *2*.

## Identification of splice junctions

RNA-seq results were analyzed using TopHat v2.1.1 (*Kim et al., 2013*; *Trapnell et al., 2009*) with no coverage search, a minimum intron size of 15 bp, and STAR v2.5.3a (*Dobin et al., 2013*) with default parameters. Splice junctions were chosen for the final annotations if they score 20 or higher in both STAR and TopHat, and if the intron length was less than 3.5 Kb (to filter out artificial splice junctions between the viral repeat regions). We also included splice junctions that were detected and were previously known but did not pass the threshold (five junctions in HHV-6A and five in HHV-6B). Two additional previously annotated HHV-6B splice junctions that were not detected were added to the final list.

## Prediction of translation initiation sites

Translation initiation sites were predicted as previously described (*Ingolia et al., 2011*; *Stern-Ginossar et al., 2012*). Briefly, a support vector machine model was trained to identify initiation sites based on normalized footprint profiles of the CHX, Harr and LTM samples (one sample of each type for each virus). A positive example set was composed of previously annotated translation initiation sites that were also well translated in our data (at least seven read counts in the normalized Harr peak, 39/58 ORFs for HHV-6A and 31/47 for HHV-6B). 10 negative examples were computed for each positive example. 2/3 of the combined set of positive and negative examples was used as a training set for the prediction model. The model was trained using a radial basis kernel, $\gamma = 2$, $C = 50$, relative positive example weighing of 1.0, and without iterative removal and retraining, and used to produce a score for each potential translation initiation site based on their CHX, Harr and LTM footprint profiles. Initiation sites that scored less than 0.5 were discarded. The remaining 1/3 of the example set was used for cross-validation, which showed 37% and 25% false-negative rate, and 2% and 5% false-positive rate for HHV-6A and HHV-6B respectively. The trained classifier was then applied to all plus and minus strand codons that had at least seven normalized Harr read counts. ORFs were then defined by extending each initiating codon to the next in-frame stop codon, and incorporating any intervening splice junctions. Previously annotated ORFs that were not recognized

by the trained model but presented observable translation in manual inspection were added to the final ORF list (*Supplementary file 3*).

Comparison of uORF and iORF conservation and kinetics uORFs were curated by selecting all ORFs of the predicted ORFs initiating in the 200 bp region upstream of each previously annotated ORF that are shorter than 200 bp. iORFs were curated by selecting ORFs longer than 20aa initiating within each previously annotated ORF. The total number of iORFs and uORFs for each main ORF was summed.

The comparison of HHV-6 annotations to HCMV was based on previously published Ribo-seq, RNA-seq and annotations of HCMV merlin strain (*Stern-Ginossar et al., 2012*; *Tirosh et al., 2015*). Text-book published lists were used to identify HHV-6 and HCMV homolog ORFs, as well as to determine the kinetic classes for previously annotated HHV-6 ORFs (*Yamanishi et al., 2013*). HCMV kinetic class annotations were taken from a proteomics-based publication (*Weekes et al., 2014*). Data for murine CMV lncRNA7.2 expression is from Tai-Schmiedel et al. unpublished.

All plot and statistical tests were done using R 3.6.0 (*R Development Core Team, 2019*; *Wickham, 2016*) on Rstudio (*RStudio Team, 2015*).

## Real-time PCR

Total RNA was isolated from a duplicate of 500,000 cells infected for 72 hr using Tri-Reagent (Sigma-Aldrich). Reverse transcription was performed with qScript Flex cDNA kit (Quantabio), using either oligo-dT or random primers, as described for each sample. Real-time PCR was performed using the SYBR Green master-mix (ABI) on a real-time PCR system StepOnePlus (life technologies), with the following primers:

lncRNA3-6A F AAAAGGACAAGAGCAGCCGC
lncRNA3-6A R ACTCGTATCACCTACCTCTCTAC
lncRNA3-6A F GGTATCGGGGTAAGAATAAGATGACG
lncRNA3-6A R AAAAGGACAAGAGCAGCCGC
lncRNA2-6B F CAAAACGGTCTCACTGCTCC
lncRNA2-6B R TCTATAAAGTGCCGTGAGTGC
lncRNA2-6A F CGACAAAACAAAATAGTCCCACT
lncRNA2-6A R ATGGAAAAGGTGGTCGTGGA
U21-6B F CCGCACCCATGAACATAAGG
U21-6B R ATGATGTGACGTGGGGACTT
U21-6A F CCAGCCACCTAGAGAACGAA
U21-6A R TTGGGCTGAACTCTCGACAT
18 S F CTCAACACGGGAAACCTCAC
18 S R CGCTCCACCAACTAAGAACG

Technical triplicate results in CT were averaged and normalized to the U21 for sample virus and to oligo-dT cDNA for each duplicate.

## Acknowledgements

We thank the members of the Stern-Ginossar lab for critical reading of the manuscript. This research was supported by the ICORE (Chromatin and RNA Gene Regulation, NS-G) and the Israeli Science Foundation (1526/18, NS-G). NS-G is incumbent of the Skirball career development chair in new scientists.

## Additional information

### Funding

| Funder | Grant reference number | Author |
|---|---|---|
| Israeli Centers for Research Excellence | Chromatin and RNA Gene Regulation | Noam Stern-Ginossar |
| Israel Science Foundation | 1526/18 | Noam Stern-Ginossar |

The funders had no role in study design, data collection and interpretation, or the decision to submit the work for publication.

### Author contributions
Yaara Finkel, Conceptualization, Software, Formal analysis, Investigation, Visualization, Writing—original draft, Writing—review and editing; Dominik Schmiedel, Conceptualization, Investigation, Writing—review and editing, Performed ribosome profiling experiments; Julie Tai-Schmiedel, Conceptualization, Investigation, Visualization, Writing—review and editing, Performed ribosome profiling experiments; Aharon Nachshon, Formal analysis, Visualization, Writing—review and editing; Roni Winkler, Investigation, Writing—review and editing, Performed ribosome profiling experiments; Martina Dobesova, Writing—review and editing, Performed ribosome profiling experiments; Michal Schwartz, Conceptualization, Writing—review and editing; Ofer Mandelboim, Conceptualization, Resources, Writing—review and editing; Noam Stern-Ginossar, Conceptualization, Supervision, Funding acquisition, Writing—original draft, Writing—review and editing

### Author ORCIDs
Yaara Finkel  https://orcid.org/0000-0002-3843-2357
Dominik Schmiedel  https://orcid.org/0000-0003-3384-5651
Michal Schwartz  http://orcid.org/0000-0001-5442-0201
Ofer Mandelboim  http://orcid.org/0000-0002-9354-1855
Noam Stern-Ginossar  https://orcid.org/0000-0003-3583-5932

### Decision letter and Author response
Decision letter https://doi.org/10.7554/eLife.50960.sa1
Author response https://doi.org/10.7554/eLife.50960.sa2

## Additional files

### Supplementary files
• Supplementary file 1. Mismatches between RNA-seq data and the HHV-6A GS reference genome.

• Supplementary file 2. Mismatches between RNA-seq data and the HHV-6B Z29 reference genome.

• Supplementary file 3. Previously annotated ORFs added manually to final ORF predictions.

• Supplementary file 4. Updated ORF annotations HHV-6A. Bed format file of genomic loci of ORFs in the genome of HHV-6A curated using SVM model predictions with manual modifications, see Materials and methods.

• Supplementary file 5. Bed format file of genomic loci of ORFs in the genome of HHV-6B curated using SVM model predictions with manual modifications, see Materials and methods.

• Supplementary file 6. lncRNA annotations HHV-6A. Bed format file of genomic loci of newly identified lncRNAs in the genome of HHV-6A.

• Supplementary file 7. Bed format file of genomic loci of newly identified lncRNAs in the genome of HHV-6B.

• Supplementary file 8. GenBank format annotation file HHV-6A. GenBank files containing annotations of ORFs, lncRNAs and splice junctions as described in this paper for HHV-6A.

• Supplementary file 9. GenBank files containing annotations of ORFs, lncRNAs and splice junctions as described in this paper for HHV-6B.

• Transparent reporting form

### Data availability
Sequencing data have been deposited in GEO under accession code GSE135363.

The following dataset was generated:

| Author(s) | Year | Dataset title | Dataset URL | Database and Identifier |
|---|---|---|---|---|
| Finkel Y, Schmiedel D, Tai-Schmiedel J, Nachshon A, Schwartz M, Mandelboim O, Stern-Ginossar N | 2019 | Comprehensive Annotations of Human Herpesvirus 6A and 6B Genomes Reveals Novel and Conserved Genomic Features | https://www.ncbi.nlm.nih.gov/geo/query/acc.cgi?acc=GSE135363 | NCBI Gene Expression Omnibus, GSE135363 |

The following previously published dataset was used:

| Author(s) | Year | Dataset title | Dataset URL | Database and Identifier |
|---|---|---|---|---|
| Shitrit A, Stern-Ginossar N | 2015 | The transcription and translation landscapes during human cytomegalovirus infection reveal novel host-pathogen interactions | http://www.ncbi.nlm.nih.gov/geo/query/acc.cgi?acc=GSE69906 | NCBI Gene Expression Omnibus, GSE69906 |

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
