## [Decision Letter]

**Acceptance summary:**

This article greatly improves our understanding of the human betaherpesvirus subfamily of the large family of herpesviruses by mapping the genomes of Human Herpesviruses 6A and 6B in great detail. By using ribosome profiling and RNAseq, several new open reading frames (ORFs) were identified, and comparison of the data with human cytomegalovirus showed that a number of these ORFs are conserved between these two betaherpesviruses. In addition, three highly abundant long non-coding RNAs (lncRNAs) have been identified. This work provides an important basis for future work on these herpesviruses.

**Decision letter after peer review:**

Thank you for submitting your article "Comprehensive annotations of human herpesvirus 6A and 6B genomes reveal novel and conserved genomic features" for consideration by *eLife*. Your article has been reviewed by three peer reviewers, one of whom is a member of our Board of Reviewing Editors, and the evaluation has been overseen by Päivi Ojala as the Senior Editor. The following individual involved in review of your submission has agreed to reveal their identity: Bhupesh K Prusty (Reviewer #3).

The reviewers have discussed the reviews with one another and the Reviewing Editor has drafted this decision to help you prepare a revised submission.

Summary:

In the manuscript "Comprehensive annotations of Human Herpesvirus 6A and 6B genomes reveal novel and conserved genomic features", Finkel et al. employed an integrative transcription and translation start site profiling approach to comprehensively characterize the gene products of HHV-6A and HHV-6B in HSB2 and MOLT-3 cells, respectively. They identified hundreds of novel viral transcripts and ORFs, including many uORFs and iORFs. The authors observed conserved patterns of uORF translation across betaherpesviruses. Supporting the role of alternate transcription start sites observed in other herpesviruses, they also observed prevalent non-canonical translation start site usage in both HHV-6A and HHV-6B. Most interestingly, they identified three highly expressed viral large non-coding RNAs that presumably reflect conserved viral large non-coding RNAs known to be expressed by other betaherpesviruses. One of these generated a non-polyadenylated stable intron that seems to be conserved among betaherpesviruses. In summary, this is first systematic study that comprehensively analyses viral gene expression in both HHV-6A and HHV-6B as well as its conservation in HCMV. The respective data will be invaluable not only for the entire HHV-6 community but for the herpesvirus field in general.

Essential revisions:

1) In the manuscript, the authors emphasize the comprehensive annotation of human herpesvirus 6A and 6B. Unfortunately, they do not provide a format the research community could use. They should provide a fully annotated GenBank file based on their exciting data (incl. new genes, splice variant and SNPs) that the community can use and cite.

2) The authors used lactimidomycin (LTM) and Harringtonine (Harr) treatments for mapping translation initiation. In Figure 1 and several other panels they show the LTM data; however, for many other panels they left it out. Was the LTM data not consistent with the Harr data in these analyses? If the LTM data was comparable to Harr, the authors should include this line as done for the other panels. If not, the authors should discuss the differences between the two treatments.

3) The authors should include a figure that depicts the signal-to-noise ratio of their Ribo-seq data (triplet-shifts of the translating ribosomes (cumulative reads in the three different frames in relation to the transcription start site)).

4) The authors identified three novel large non-coding RNAs, two of which are heavily spliced. The transcription start site profiling RNA-seq data are very convincing regarding the existence of these viral lncRNAs. As these lncRNAs appear to be highly expressed, it should be relatively straight forward to confirm them by Northern Blot. In particular, this would reveal alternatively spliced isoforms which may differ in function. Moreover, this would provide a much more solid basis of these fascinating novel gene products for future studies.

5) The continuous increase in viral gene expression throughout lytic infection results in a much broader range of transcripts (transcription start sites) late in infection. This will result in an increased number of uORFs and iORFs late in infection. It is interesting that viral late genes tend to have more uORFs than early genes. How reliable is the current annotation of viral early and late genes? The authors imply that uORFs and iORFs originate from alterations in the fidelity of the ribosomes at late times of infection. However, early viral mRNAs are still transcribed, present and translated at late times of infection. Therefore, it is unclear why viral late genes should be preferentially affected regarding translation of viral uORFs. Could the observed correlation simply result from a higher level of gene expression of the viral late genes compared to viral early genes (and thus a higher sensitivity for picking up uORF translation)? The authors should assess the impact of gene expression levels on their ability to detect viral uORFs and how this relates to their observation that uORFs are more prevalent in viral late genes.

6) Similarly, iORFs will become more prevalent throughout lytic infection due to the increase in viral gene expression and the increase in the number of different viral transcripts present within the infected cells at late times of infection. While HHV6 iORFs may result from a failure of translation initiation at the upstream AUG, it is equally if not more likely that they originate from alternative transcripts that only initiate downstream of the respective AUG. This was most prevalent for HSV-1. While the RNA-seq approach the authors employed provides a strong transcription start site enrichment, it is not nearly as sensitive (transcription start site enrichment) as other approaches, e.g. dRNA-seq. Therefore, the authors are likely to have missed a decent number of transcription start sites that presumably explain translation of the viral iORFs they observed. This is particularly likely for highly expressed genes, which will mask embedded alternative transcription start sites. The authors should carefully rephrase their discussion to also consider more promiscuous transcription initiation (rather than translational fidelity) at late times of infection explaining novel iORFs and uORFs.

7) The non-polyadenylated lncRNA3 is particularly interesting as a small non-coding RNA (U77) has previously been reported to be expressed from this region by Nukui, Mori and Murphy, 2015. Moreover, Prusty et al., 2018a, also demonstrated induction of viral transcription from this region immediately after viral reactivation. The authors may want to discuss their results in the light of these previous reports.

8) Please provide a figure depicting the use of non-canonical start sites during the 3 different stages of infection for HHV-6B.

9) Please provide more details how sequencing data were processed from the viral repeat regions. Currently, no data is shown from ORFs transcribed from the DR regions.

---

## [Author Response]

Essential revisions:1) In the manuscript, the authors emphasize the comprehensive annotation of human herpesvirus 6A and 6B. Unfortunately, they do not provide a format the research community could use. They should provide a fully annotated GenBank file based on their exciting data (incl. new genes, splice variant and SNPs) that the community can use and cite.

As requested by the reviewers, in addition to the separate BED files containing lncRNA and ORF annotations, and the CSV files with the splice junctions (that was provided and deposited with our original submission), we now provide a combined GenBank file for each virus, as Supplementary files 8 and 9. We did not include SNPs analysis in this GenBank file because we cannot be confident based on our experiments that the mismatches identified can be generalized beyond the lab strain we have worked with.

2) The authors used lactimidomycin (LTM) and Harringtonine (Harr) treatments for mapping translation initiation. In Figure 1 and several other panels they show the LTM data; however, for many other panels they left it out. Was the LTM data not consistent with the Harr data in these analyses? If the LTM data was comparable to Harr, the authors should include this line as done for the other panels. If not, the authors should discuss the differences between the two treatments.

In our original submission in the figures in which the LTM data was consistent with the Harr data we chose to show only the Harr data (since we wanted to make the figures a bit less dense). However, as requested by the reviewers and to enhance clarity we now added the LTM tracks to all the main figures (Figures 2, 4 and 8).

3) The authors should include a figure that depicts the signal-to-noise ratio of their Ribo-seq data (triplet-shifts of the translating ribosomes (cumulative reads in the three different frames in relation to the transcription start site)).

As requested by the reviewer we now added data that shows the P-site mapping of reads in all three frames. This figure is not dramatically different from the analysis that we already presented in Figure 6D and we now added it as Figure 1—figure supplement 1. Both analyses show that around 50% of our footprints data maps to frame 0. Since the RNAse-I digestion of the ribosome-protected fragments is imperfect, each individual footprint provides a limited statistical evidence of the ribosome position. However, averaging multiple reads allows unambiguous determination of the reading frame (in sharp contrast to the 5′ termini of the mRNA reads). The observed level of enrichment in footprints that align to the translated frame is analogous to that observed in previous mammalian cells libraries (Stern-Ginossar et al., 2012) and indicate the ribosome profiling data is of high quality. It is important to note this analysis only reflects the signal-to-noise of our P-site mapping but not the signal-to-noise of our ORF predictions as these predictions are not dependent per-se on our single-base resolution mapping as the SVM is trained on the Harr, LTM and CHX ribosome density profiles of a set of well-expressed canonical viral ORFs. Therefore, the signal to noise of the predictions is calculated by false positive and false negative rates of a cross-validation set (2% and 37% for HHV-6A and 5% and 25% for HHV-6B, respectively), as detailed in the Materials and methods section. This means that we are likely missing expressed ORFs in our annotations but our false discovery rate is relatively low.

4) The authors identified three novel large non-coding RNAs, two of which are heavily spliced. The transcription start site profiling RNA-seq data are very convincing regarding the existence of these viral lncRNAs. As these lncRNAs appear to be highly expressed, it should be relatively straight forward to confirm them by Northern Blot. In particular, this would reveal alternatively spliced isoforms which may differ in function. Moreover, this would provide a much more solid basis of these fascinating novel gene products for future studies.

As requested by the reviewers we attempted to provide evidence for hvv6 encoded lncRNAs expression using Northern blot analysis. Since we were not able to obtain a permit for radioactive work in the time-frame for revisions, we have used biotin labeled probes which might be less sensitive. With these, we were able to detect the lncRNA3 intron band at the expected size, but not the shorter and less abundant lncRNAs. Since the reviewers raised the point of relative abundance of the different lncRNA3 isoforms we thought an important issue to address is the true abundance of lncRNA3 intron (which does not have poly-A tail) relative to its spliced poly-adenylated isoforms and the rest of the viral transcripts. We therefore conducted an additional RNA-seq experiment, this time without poly-A enrichment step. This experiment revealed that the intron of HHV-6B lncRNA3 is more than 100-fold more abundant than the spliced lncRNA3 isoforms. We also performed a similar experiment in HCMV infected cells in which we found that the HCMV RNA5.0 intron is 10-fold more abundant than spliced the RNA5.0. While we were not able to probe for the relative expression of the different spliced lncRNA3 isoforms, we think the extremely high abundance of the intron, the uniqueness of a stable intron feature and its conservation in different β herpesviruses points that the intron itself is probably the dominant functional element in this locus. It is therefore possible that the variety of 5’ TSSs and splicing isoforms in this locus, all sharing the same intron, arise from a weaker selection on other features of this locus. This option is now further discussed in the fifth paragraph of the Discussion.

The Northern blot analysis and RNA-seq results were added as Figure 5—figure supplement 1.

5) The continuous increase in viral gene expression throughout lytic infection results in a much broader range of transcripts (transcription start sites) late in infection. This will result in an increased number of uORFs and iORFs late in infection. It is interesting that viral late genes tend to have more uORFs than early genes. How reliable is the current annotation of viral early and late genes? The authors imply that uORFs and iORFs originate from alterations in the fidelity of the ribosomes at late times of infection. However, early viral mRNAs are still transcribed, present and translated at late times of infection. Therefore, it is unclear why viral late genes should be preferentially affected regarding translation of viral uORFs. Could the observed correlation simply result from a higher level of gene expression of the viral late genes compared to viral early genes (and thus a higher sensitivity for picking up uORF translation)? The authors should assess the impact of gene expression levels on their ability to detect viral uORFs and how this relates to their observation that uORFs are more prevalent in viral late genes.

The reviewers are correct and at late time points it is likely there are more cryptic transcription start sites, which will result in more uORFs and iORFs translation (this is now more explicitly explained in the third paragraph of the Discussion). However, this should be true for both late and early genes that are expressed at late stages of infection. As pointed out by the reviewers, higher expression of a gene at late time points of infection increases the probability for detecting novel uORFs and iORFs related to it. Nonetheless, our data reveals that late genes have relatively more uORFs but not more iORFs. This distinction supports the assumption that there is probably a specific enrichment of uORFs in late genes that does not only stem from differences in their expression. To further explore this issue, we tested for correlation between RNA expression level of each ORF and the number of related uORFs found. As may be expected, some correlation (Pearson's R = 0.34) was found in HHV-6B ORFs, but not in HHV-6A or HCMV ORFs (R = 0.04 and R = 0.03 respectively), suggesting expression levels probably do not solely explain the enrichment we see for uORFs in late genes (this analysis is now presented in Figure 7—figure supplement 3). Also, it is worth noting that we are not arguing late genes have intrinsic higher propensity to recruit ribosomes, rather, that late genes probably tend to acquire 5’ UTRs features that support more uORF initiation (i.e. near cognate codons in the right distance and the right Kozak context). As for the reliability of current kinetic class annotation, we present in Figure 7A (and Figure 7—source data 1) the classification of HHV-6B ORFs together with the temporal clusters that are based on our Ribo-seq measurements. As we describe in the Results, the current annotation agrees quite well with our measurements, with few exceptions, some of which are discussed in the text.

6) Similarly, iORFs will become more prevalent throughout lytic infection due to the increase in viral gene expression and the increase in the number of different viral transcripts present within the infected cells at late times of infection. While HHV6 iORFs may result from a failure of translation initiation at the upstream AUG, it is equally if not more likely that they originate from alternative transcripts that only initiate downstream of the respective AUG. This was most prevalent for HSV-1. While the RNA-seq approach the authors employed provides a strong transcription start site enrichment, it is not nearly as sensitive (transcription start site enrichment) as other approaches, e.g. dRNA-seq. Therefore, the authors are likely to have missed a decent number of transcription start sites that presumably explain translation of the viral iORFs they observed. This is particularly likely for highly expressed genes, which will mask embedded alternative transcription start sites. The authors should carefully rephrase their discussion to also consider more promiscuous transcription initiation (rather than translational fidelity) at late times of infection explaining novel iORFs and uORFs.

We never intended to imply that iORF initiation are likely stemming from changes in the ribosome fidelity and we fully agree that probably many of the alternative translation products we see stem from alternative transcription initiation (which was also the main feature that stemmed from our previous HCMV publication). This is now more explicitly stated in the Discussion (Discussion, third paragraph). A representative example for that can be found in the HHV-6 gene U53 and the HCMV gene UL80, as seen in Figure 7B of our manuscript.

7) The non-polyadenylated lncRNA3 is particularly interesting as a small non-coding RNA (U77) has previously been reported to be expressed from this region by Nukui, Mori and Murphy, 2015. Moreover, Prusty et al., 2018a, also demonstrated induction of viral transcription from this region immediately after viral reactivation. The authors may want to discuss their results in the light of these previous reports.

We thank the reviewers for bringing these results to our attention. We now discuss these results as follows: "Interestingly, a small non-coding RNA (sncRNA-U77) that is mapped to the intron of lncRNA3 was shown to be expressed by HHV-6A (Nukui, Mori and Murphy, 2015). It is therefore possible that the stable intron is further processed to create different functional elements. Furthermore, a recent study showed that TSA-mediated HHV-6A transactivation results in increased transcription from a region overlapping the lncRNA3 locus (Prusty et al., 2018a), implying lncRNA3 may be involved in HHV-6A reactivation. "

8) Please provide a figure depicting the use of non-canonical start sites during the 3 different stages of infection for HHV-6B.

To address this point, we added Figure 7—figure supplement 4, which is referred to in the subsection “uORFs are enriched in betaherpesvirus late genes”. This figure shows a higher proportion of non-AUG start codons in ORFs that are expressed in late kinetics according to our clustering analysis. This increased use of non-canonical start codons at late time points may further point on potential changes in the translation apparatus.

9) Please provide more details how sequencing data were processed from the viral repeat regions. Currently, no data is shown from ORFs transcribed from the DR regions.

In our analysis, we ignored reads that were aligned to multiple locations in the genome therefore the repeat regions were not included in our analysis. We now clearly state this in the Materials and methods section: "Reads aligned to multiple locations were discarded, therefore, repeat regions of the genome were not included in our analysis".